# The 2025 Starfish Barometer

Marina Lévy[1], Karina von Schuckmann[2], Patrick Vincent[2], Bruno Blanke[3], Joachim Claudet[4], Patrice Guillotreau[5], Audrey Hasson[2], Claire Jolly[6], Yunne Shin[5], Olivier Thébaud[7], Adrien Vincent[8], and Pierre Bahurel[2]

[1]Sorbonne Université, LOCEAN-IPSL, CNRS/IRD/MNHN, Paris, France
[2]Mercator Ocean International, Toulouse, France
[3]LPO-IUEM, UBO/CNRS/Ifremer/IRD, Brest, France
[4]CNRS, PSL-EPHE-UPVD, CRIOBE, Paris, France
[5]MARBEC, IRD/Ifremer/CNRS/Université Montpellier, Montpellier, France
[6]OECD, Paris, France
[7]AMURE, Ifremer/UBO/CNRS/IRD, IUEM, Plouzané, Brest, France
[8]IPOS, Paris, France

**Correspondence:** Marina Lévy (marina.levy@locean.ipsl.fr)

**Abstract.** The Ocean is essential to life on Earth, regulates the climate, supports rich biodiversity, sustains livelihoods, and inspires cultures and societies. However, unregulated human impacts are putting the Ocean and its ability to contribute to humanity at risk. The Starfish Barometer is a new initiative launched on World Ocean Day (8 June 2025) to provide a concise, science-based annual overview of the multiple dimensions of the Ocean through the lens of its interdependence with humanity.

Each year, the Starfish Barometer will present a carefully curated selection of ocean-related developments, chosen for their global significance and grounded in the most up-to-date scientific evidence, intended for a broad non-specialist audience. Rather than offering an exhaustive review, it will spotlight key aspects, robust, evidence-based, and reflective of major developments of the year. The Starfish Barometer emphasizes the two-way relationship between humanity and the Ocean: we impact its future, and it shapes ours. Its distinctive format, a five-armed starfish with the current state of the Ocean on the top arm,

visually represents the balance conveyed by the four remaining arms: human-induced pressures that are undermining ocean health ; the resulting harms to society ; the ocean protection efforts underway ; and the opportunities that the Ocean continues to offer to humanity. Key figures from the 2025 edition illustrate the urgent state of the Ocean. Sea level has risen by 23 cm since 1901; in 2023 alone, the losses from tropical storms and flooding totaled US$ 102 billion. 2024 ocean temperatures break the 64-year record, with sea-surface temperature and marine heatwaves showing a marked increase globally. Marine animal

food production reached a record 115 million tonnes in 2022, yet 37.7% of fish stocks remain overexploited, highlighting the urgent need for sustainable practices. Declared protection policies currently cover 8.34% of the Ocean's surface, while marine biodiversity is under threat, with now 1,677 marine species recorded as at risk of extinction.

## 1 Introduction

The Ocean is essential to life on Earth — it regulates the climate, supports rich biodiversity, sustains livelihoods, and shapes

cultures. Yet, information about its current state remains fragmented across disciplines, reports, and institutions, often inac-

cessible to the general public. The Starfish Barometer responds to a pressing need: to bring together, once a year and in one place, the most recent, reliable, and evidence-based scientific knowledge about the Ocean for the general public. Published each year on World Ocean Day (June 8), the Starfish Barometer serves as a yearly civic rendezvous by translating the year's most significant ocean-related developments into accessible science-based insights.

What makes the Starfish Barometer unique is not the novelty of the data it presents — on the contrary, all the information it includes has already been published and validated in scientific literature, institutional or international reports. Its strength lies in how it brings together this dispersed knowledge, synthesizing it into a comprehensive and integrated overview of the Ocean, aligned with the United Nations Sustainable Development Goal related to the Ocean SDG 14 (UN, 2015). The Barometer is designed to serve a broad non-specialist audience, including policy makers, educators, civil society actors, and the general

public, while remaining grounded in peer-reviewed science to ensure credibility and relevance for the scientific community as well. The Starfish Barometer takes stock of the latest available information on historical changes, current status and trends, rather than focusing on projected futures. In this sense, it acts as a snapshot of the year, drawing attention to key developments and current trends, or highlighting important knowledge gaps. Each year, the Starfish Barometer selects and curates ocean-related developments — such as new or updated scientific findings, international policy decisions, or governance milestones —

chosen for their global relevance and based on the most recent knowledge available at the time of publication. This selection is not exhaustive, but reflects key signals from the year, robust, factual, and representative of major trends.

The Starfish Barometer adopts a distinctive perspective, looking at the Ocean through the lens of its interdependence with humanity. Humans do both good and harm to the Ocean — and the Ocean, in turn, does both good and harm to humans. By presenting this dual relationship in a holistic and balanced way, the Starfish Barometer highlights the complex web of interactions

that shapes the Ocean-humanity relationships. It puts into perspective human pressures and protection efforts, societal harms, and opportunities for humanity, offering a clear and accessible overview of these dynamics to empower informed decisions and commitments for ocean protection. This approach emphasizes that the Ocean is a vital environment from which we derive many benefits — provided we take good care of it.

To visualize the balance in the relationships between humanity and the Ocean, the Starfish Barometer takes the shape of

a five-armed starfish (Figure 1). The top arm offers a global view of the Ocean's state. The two arms on the left side of the starfish represent negative developments: human-induced pressures and resulting harms to society; the two arms on the right side reflect positive dynamics: efforts to protect the Ocean and opportunities the Ocean provides to humanity. The upper arms show humanity's negative and positive impacts, while the lower arms show how the Ocean harms and creates opportunities for humanity. Thus the Starfish can be read horizontally, from left to right (negative to positive), or vertically, from top to

bottom (human action to societal consequence). Selected key informations are featured as concise, evidence-based news items, arranged along each branch of the Starfish Barometer and highlighted through short, clear, and accessible headlines. This format ensures scientific objectivity while making information accessible to a broad audience. The present article establishes the peer-reviewed scientific foundation of the Starfish Barometer (www.starfishbarometer.org).

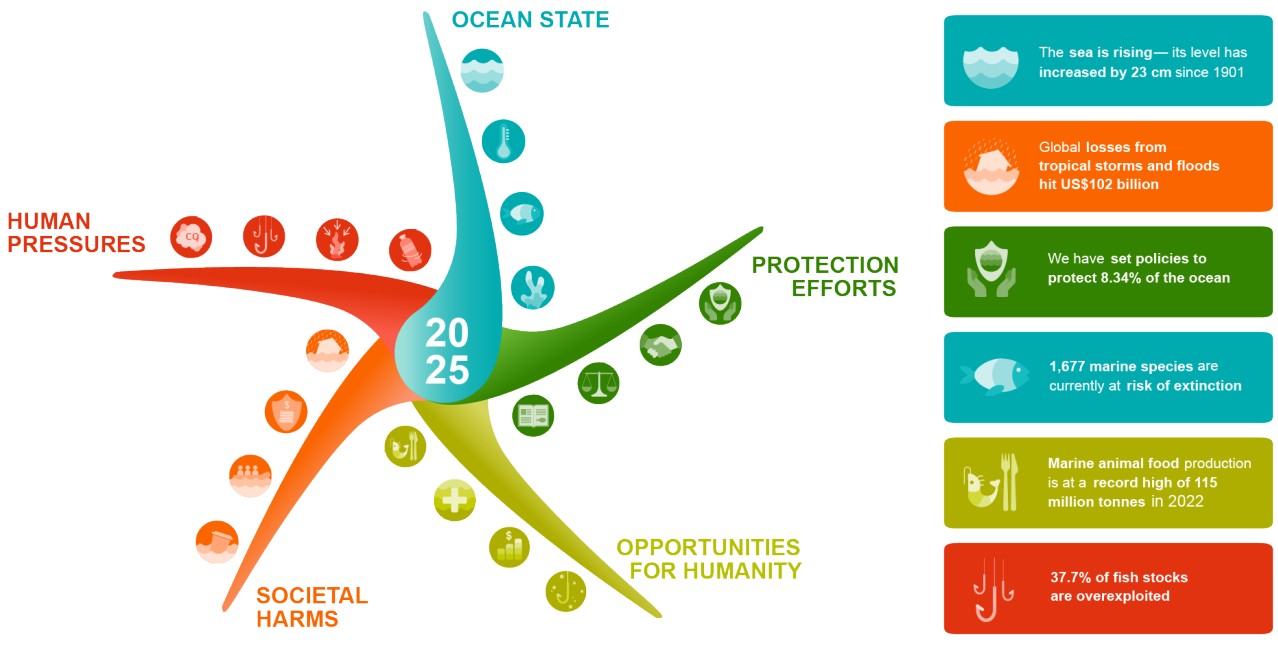

**Figure 1.** Schematic of the five branches of the Starfish Barometer and key figures for 2025. At the top, the blue branch represents the current state of the Ocean. The four other branches can be read horizontally, from red tones on the left to green tones on the right (negative to positive), or vertically, from darker tones at the top to lighter tones at the bottom (human actions to societal consequences). The figures represent the most recent estimates available on 8 June 2025.

## 2 Methodology

The content of the Starfish Barometer was curated by a multidisciplinary group of experts, who identified and selected significant recent developments to be featured in the five thematic branches. The process began with each expert proposing candidate news items, based on a set of common criteria: global relevance, robustness of the underlying data, significance of the development, and relevance to the Ocean-human relationship. Sources included scientific assessments (e.g. IPCC, FAO, OECD), new or updated datasets (e.g. Global Carbon Budget), recent peer-reviewed scientific publications, international policy de-

velopments, data gaps, and emerging alerts. The proposed items were then collectively reviewed and discussed by the expert group to determine their relevance and final selection. This process is qualitative and adaptive by design, guided by transparent criteria and expert judgment to ensure that each Barometer edition remains accessible to the general public and focused on widely relevant issues.

The allocation of news items to specific branches is guided by thematic relevance but is not exclusive, as many ocean-

related developments intersect multiple dimensions. For example, certain topics, such as illegal, unreported and unregulated

(IUU) fishing, could be considered both a human pressure and a threat to societal equity. In such cases, items were assigned to the branch most closely aligned with the primary focus of the issue, following a consistent editorial framework rather than rigid topic boundaries. Each branch was limited to one page and to four key messages, anchored in recent developments and supported, in most cases, by figures and trend analyses.

We intentionally chose not to establish explicit logical connections between the news items within or across branches. While the branches together form a coherent picture of the Ocean-human relationship, forcing a linear or causal narrative might create an illusion of completeness or determinism that does not reflect the complexity of ocean issues. Instead, each message stands on its own as a robust signal from the past year, contributing to a broader mosaic of understanding.

All figures presented in the Starfish Barometer are drawn directly from authoritative sources and are reported as published, in order to ensure transparency and traceability. Readers interested in the methodological details or associated uncertainties are encouraged to consult the original sources.

Each annual edition of the Starfish Barometer will follow the same symbolic and structural framework, with its five same thematic arms : Ocean state, human pressures, societal harms, protection efforts and opportunities for humanity. However, the content within each arm will evolve yearly to reflect the most recent and relevant developments. When new data become available — such as for carbon emissions or sea level — they will be updated accordingly. Yet not all information is refreshed annually, and the Barometer's non-exhaustive approach provides the necessary flexibility to highlight either new figures or important issues that may vary from one year to the next. In this sense, the Barometer differs from initiatives like the Global Carbon Project (Friedlingstein et al., 2025) or the Ocean health index (Halpern et al., 2012): rather than systematically updating a fixed dataset, it offers a curated, narrative-based selection that captures the most meaningful signals of the year.

## 3    2025 Starfish five arms

### 3.1    Ocean State

**The sea is rising – its level has increased globally by 23 cm since 1901**, as a direct consequence of accelerated ocean warming and increasing ice loss from ice sheets and glaciers. In 2024, global sea levels reached the highest level ever recorded since monitoring began, rising 23 cm since 1901  (WMO, 2025; Forster et al., 2025). Over the past decade (2015–2024), the rate of sea-level rise has been twice as fast as in the early years of satellite records (1993–2002), due to increasing ice loss from the Greenland and Antarctic ice sheets, ongoing glacier mass loss, and thermal expansion due to ocean warming (WMO, 2025).

**The Ocean reached its highest recorded temperatures**, with oxygen loss and rapid acidification threatening marine life. 2024 marks a year of intense, persistent and widespread heatwaves in the Ocean, and ocean heat content reached its highest level in the 64 years for which we have reliable recorded global observations (since 1960), surpassing the previous record high set in 2023 (WMO, 2025; Cheng et al., 2025; Pan et al., 2025), and the 2015/16 record by 0.25°C at the ocean surface (Terhaar et al., 2025). The record ocean surface temperature values reflect natural variability amplified by long-term global warming — an event unlikely to occur without the underlying climate trend (Terhaar et al., 2025; Guinaldo et al., 2023).

Ocean warming has doubled over the past two decades and has been accelerating for the past 70 years (IPCC, 2019; von Schuckmann et al., 2023; Minière et al., 2023; Storto and Yang, 2024; Forster et al., 2024). Ocean surface acidity (in $[H^+]$ concentration) has risen by 30% over the past 39 years (IPCC, 2021; von Schuckmann et al., 2024; WMO, 2025), and open ocean waters have experienced an oxygen reduction of 0.8 to 2.4% over the past half-century to century (Breitburg et al., 2018; Oschlies, 2021; IPCC, 2021).

**1,677 marine species are threatened with extinction**, with a third of sharks and over a quarter of cetaceans critically endangered, mainly due to overfishing and climate change. As of the most recent update (May 2025), 1,677 marine species are listed as at risk of extinction on the IUCN Red List, including 291 classified as critically endangered, 647 as endangered, and 739 as vulnerable (IUCN, 2025). Among these, 1,211 species are experiencing declining populations, 28 are increasing, and the remainder are either stable or have unknown trends. This represents an increase of 204 species at risk since the last reported estimate (IOC-UNESCO, 2024). One third of sharks, rays, and chimaeras are classified as threatened, with 67% at risk of extinction due to fisheries (Jabado et al., 2024). 26% of 92 cetacean species (whales, dolphins, and porpoises) are threatened with extinction, and 11% are near-threatened. The percentage of threatened cetaceans has increased over time, from 15% in 1991 to 26% in 2021 (Braulik et al., 2023). The Ocean Census initiative has discovered 866 new marine species in its first year, highlighting the unknown depths of ocean life. This pace of discovery highlights a critical risk: countless marine species may vanish before science even has a chance to identify them, particularly in the deep-sea where multiple stressors such as climate change, deep sea mining and bottom trawling puts them at risk (IPOS, 2025), making conservation efforts all the more urgent (Ocean Census, 2025).

**The fourth major coral bleaching event recorded hit the Ocean**, with almost half of all coral species threatened with extinction and a worrying acceleration in reef degradation. Following extreme conditions in 2023 driven by historically high heat stress, the 2024 coral bleaching event is the fourth global event on record since 1985 and the second in the past decade (Hoegh-Guldberg et al., 2023; NOAA, 2024). Live coral cover on reefs has nearly halved over the past 150 years, with the decline dramatically accelerating over the past two or three decades (IPBES, 2019). A total of 44% of reef coral species are threatened with extinction (IUCN, 2025). The rapid decline in coral reefs — the Ocean's most biodiverse habitat — is weakening natural storm protection, threatening biodiversity, and endangering livelihoods (IPBES, 2019).

### 3.2 Human pressures

**Fossil fuel $CO_2$ emissions are rising, including a 2.7% increase from shipping.** The 2024 Global Carbon Budget projects global fossil carbon dioxide ($CO_2$) emissions of 37.4 billion tonnes, up 0.8% from 2023 (Friedlingstein et al., 2025). Emissions from international shipping are also on the rise, with a projected 0.6 billion tonnes of $CO_2$ in 2024 (representing 1.6% of total global emissions), up 2.7% from 2023 (Andrew et al., 2024); these figures rely on countries' self-reported data regarding fuel sales for international shipping. Fossil fuel $CO_2$ emissions are the main contributor to recent climate change (IPCC, 2021). They also lead to ocean acidification as approximately one-third of these emissions are absorbed by the Ocean (Friedlingstein et al., 2025).

**Unsustainable fishing hits 37.7%, while 75% of large vessels go untracked.** The percentage of marine stocks fished at unsustainable levels (i.e. beyond maximum sustainable yield) has increased since the mid-1970s, from 10% in 1974 to 37.7% in 2021 (FAO, 2024). Marine fisheries activities involve approximately 4.9 million registered motorized and non-motorized fishing vessels in 2022 (FAO, 2024). This estimate represents only part of the unsustainability challenge, as it does not account for broader ecological and social impacts of fisheries (Asche et al., 2025), nor for IUU fishing. An estimated 75% of large (>15m) fishing vessels operate without proper tracking (2017–2021), challenging growing efforts to combat IUU fishing and to regulate marine protected areas (MPAs) (Paolo et al., 2024).

**Rising human and climate pressures threaten marine habitats globally.** A combination of growing pressures is involved, including unsustainable fishing practices, pollution, ocean warming, acidification, and oxygen loss pushing marine ecosystems and species beyond their tolerance limits. Climate change is driving the loss of suitable habitats globally, with the most pronounced shifts in species richness occurring in tropical ecosystems (Chaudhary et al., 2021). Coastal ecosystems are also showing increasing signs of degradation. Since 1950, coastal hypoxia has increased tenfold, reducing the extent and quality of coastal habitats critical to marine biodiversity (Breitburg et al., 2018). Climate change and pollution-driven eutrophication have caused a rise in the reporting of nearshore macroalgal blooms (green tides) worldwide over the past two decades (Ren et al., 2024), although comprehensive global estimates are lacking. 50% of mangrove ecosystems are at risk of collapse (IUCN, 2024), over half of World Heritage seagrass habitats have high vulnerability to climate change (Losciale et al., 2024), while 33% of the world's sandy coastline is currently hardened by man-made structures (Nawarat et al., 2024). The multiple uses of coastal areas — like coastal infrastructures, tourism, fish farming, and offshore structures — are putting more pressure on ocean habitats, through pollution, habitat destruction, increased risks of marine pathogens, parasites and invasive species, especially as coastal populations grow: in 2018, about 2.2 billion people, or 30% of the world's population, lived within 50 km of the coast (Cosby et al., 2024).

**Plastic pollution is rising but still no global system to monitor and assess it.** Global plastic production has surged from 2 million tonnes (Mt) in 1950 to 413.8 Mt in 2023, of which only 8.7% is from recycled origin (Plastics Europe, 2024). Plastic waste accounts for over 80% of identified aquatic debris (Harris et al., 2023). In 2021, plastic accumulation in rivers and the Ocean was estimated at 75–199 Mt (UNEP, 2021). Marine plastic pollution has been reported since the 1970s, but the absence of a unified global monitoring system still hampers effective assessment and response (Galgani et al., 2024).

## 3.3 Societal harms

**Global losses from tropical storms and floods hit US$ 102 billion.** In 2023, the loss of economic assets caused by tropical cyclones and floods was worth US$ 102 billion (inflation-adjusted, Munich RE, 2025a, b). Since 1980, losses have grown exponentially and now regularly exceed US$ 100 billion per year, costing 25% more to human populations every decade in inflation-adjusted terms (Nordhaus, 2006). Tropical storms often trigger heavy rainfall resulting in floods, casualties, and destruction of assets worth billions of dollars (Kron et al., 2019). Approximately 560 million people are exposed yearly and this number has increased across all cyclone intensities over the past two decades (Jing et al., 2024).

**Insurance premium costs incurred by maritime activities grew by 5.9%** with the expansion of global trade and increasing threats. Global marine insurance premiums totaled US\$ 38.9 billion in 2023, a 5.9% increase from 2022 (IUMI, 2024). Steady growth has occurred over the past 5 years. Geopolitical tensions on global trade routes have significantly impacted insurance premiums and increased voyage costs by rerouting vessels, causing longer transit times. The growth also reveals the rise in insurance costs against hurricanes, war conflicts and piracy. The main insurance premiums reported by the International Union of Maritime Insurance (IUMI) and P&I (Protection and Indemnity) clubs insure around 95% of all risks, but public insurance figures provide only a partial view of total risk exposure (Knapp and Heij, 2017). In particular, damages to marine ecosystems in case of oil spills vary widely and some cannot be valued in monetary terms (El Moussaoui and Idelhakkar, 2023).

**9,002 migrants lost their lives at sea** — the highest casualty loss recorded in the past decade. More than 73,000 migrants have died or disappeared at sea during the past decade, with the highest toll in 2024 at 9,002 fatalities or disappearances (IOM, 2025). This represents a 3% increase from 2023, but a 25% increase from 2022. Poverty, unemployment, starvation and in-security are the main drivers of migration (Ndaliman and Abubakar, 2024). Changing environmental conditions, e.g. from sea-level rise, depletion of marine resources due to overfishing and climate variability, are also important drivers of forced displacement (McLeman et al., 2025). However, the specific relationship between oceanic changes and migrant fatalities remains unclear and would require further investigation.

**US\$ 250 billion health costs and 1,200 species affected by marine plastic pollution**. In 2015, health costs related to plastic exposure through seafood exceeded US\$ 250 billion globally (Landrigan et al., 2023). Chemicals carried by microplastics have been found in human blood, fat, and urine, and are linked to cancer, infertility, obesity, heart disease, and developmental issues in babies — even before birth (European Environment Agency et al., 2021). Over 1,200 marine species were reported to be harmed by plastic pollution through entanglement, ingestion and chemical contamination (Santos et al., 2021).

### 3.4 Protection efforts

**Global protected areas hits 8.34%, but efforts are still needed** to effectively reach the 30% target. Over the past decade, the global coverage of MPAs has increased significantly, from 3.72% of the global Ocean in 2015 (Lubchenco and Grorud-Colvert, 2015) to 8.34% in 2024 (corresponding to 30.238 millions of km$^2$), but still far from the $30 \times 30$ target (protecting 30% of the coastal and marine waters by 2030) (Protected Planet, 2025; UNEP-WCMC and IUCN, 2024). While 19% of national waters (39% of the Ocean) are protected, just 1.45% of Areas Beyond National Jurisdiction (61% of the global Ocean) have protection (Protected Planet, 2025). Between 2023 and 2024, the surface covered by MPAs increased only by 0.007% (Protected Planet, 2024). Globally, one in four MPAs exist only on paper (i.e. are not effectively implemented) and an additional third fails to truly support conservation goals (Pike et al., 2024). Only a third of the global coverage of MPAs is fully or highly protected (Pike et al., 2024). Ensuring that the global network of MPAs delivers its expected benefits for climate, biodiversity, and food security requires not only achieving the 30% coverage target, but also increasing the level of protection within MPAs (Arneth et al., 2023).

**Multilateral ocean governance advances,** highlighted by 21 ratifications of the High Seas Treaty. In 2024, the UN High Seas Treaty entered a critical phase with national ratifications and implementation planning. As of April 2025, 21 countries

had ratified the treaty (High Seas Alliance, 2025). This treaty aims to promote the conservation and sustainable use of marine biodiversity beyond national jurisdictions (BBNJ) — covering nearly two thirds of the Ocean (Blasiak and Claudet, 2024) — including via capacity building, transfer of marine technology and equitable sharing of the benefits of marine genetic resources. In 2024, new global guidelines were adopted to strengthen sustainable fisheries management by improving transparency, combating IUU fishing, restricting catch limits, and expanding AI-powered satellite monitoring (FAO, 2024; European Commission, 2024; Global Fishing Watch, 2024). 2024 also saw major advancements in blue carbon initiatives and marine renewable energy (Global Mangrove Alliance, 2024). In March 2022, a resolution was adopted by the UN to end plastic pollution through a legally binding agreement (UNEA-5). In July 2023, the International Maritime Organization adopted a new climate strategy targeting a 40% cut in carbon intensity from international shipping by 2030, alongside a push for at least 5% of the sector's energy to come from zero or near-zero emission fuels and technologies (IMO, 2023). In 2024, the International Tribunal for the Law of the Sea issued a landmark opinion (ITLOS, 2024), addressing States' obligations to protect the Ocean from climate change impacts within the framework of the United Nations Convention on the Law of the Sea (UNCLOS)."

**Equity rises as a driving force in global ocean conservation**, climate and sustainable development. Equity is increasingly recognized as an enabler of environmental sustainability, economic development, and global stability (Österblom et al., 2023; Claudet et al., 2024). The Convention on Biological Diversity (CBD) explicitly introduces equity in the Aichi Target 11 (CBD, 2010) and in the Kunming-Montreal Global Biodiversity Framework  (CBD, 2022). Equity is a core principle of the 2030 Sustainable Development Agenda (UN, 2015). Since 2021, the UN Decade of Ocean Science for Sustainable Development has supported equitable access to ocean knowledge and capacity building (Claudet et al., 2020). Equity is now central in sustainable planning efforts (Strand et al., 2024). Grassroots resistance efforts led by coastal communities have successfully stopped unfair exposure to environmental harms and preserved their livelihoods (Blythe et al., 2023).

**Ocean literacy is gaining global momentum**, with unprecedented growth in educational activities. Ocean Literacy is becoming a key pillar of Ocean and coastal management and policy (Shellock et al., 2024). Since 2021, 418 ocean literacy activities and 27 projects have been endorsed under the UN Decade of Ocean Science. In 2024, the release of the Venice declaration offers a global framework for ocean literacy (IOC-UNESCO, 2024; Ocean Literacy World Conference, 2024).

### 3.5   Opportunities for humanity

**Marine animal food production reached a record 115 million tonnes**, meeting a global demand with environmental costs. Marine aquaculture has almost quadrupled since the 1990s, reaching 35.3 Mt in 2022 (FAO, 2024), and now corresponds to 31% of the total production of marine animal foods (FAO, 2024). In comparison, global marine capture fisheries production has remained relatively stable since the 1990s, reaching 79.7 Mt in 2022. However, while sustainable aquaculture production is possible, the sector is very diverse, with some aquaculture production systems generating significant environmental impacts (Garlock et al., 2024). Overall, the global apparent consumption of aquatic animal food increased by 3% per year between 1961 and 2021 — outpacing both the growth of the world's population (1.6% per year) and the consumption of terrestrial meats (FAO, 2024). Over the same period, the internationally traded value of aquatic animal products grew by an average of 4.0% annually, highlighting their increasing importance as one of the most extensively traded food commodities

worldwide (FAO, 2024). These trends underscore that strengthening the sustainability of aquatic food production is essential to reduce environmental impacts and harness emerging economic opportunities.

**The marine-derived health market is growing amid high inequity**, with +7.5% for marine-derived pharmaceutical sales. In 2024, a landmark global database containing 308.6 million gene clusters of marine origin was published (Laiolo et al., 2024), marking the largest collection of its kind to date and highlighting the acceleration in bioprospecting for new marine genetic resources (Krusberg et al., 2024). Future potential is vast given that 70–90% of marine species are still undescribed (Sigwart et al., 2021). The global market for marine-derived pharmaceuticals was valued at US$ 4.1 billion in 2023 (Fact.MR, 2023), while the marine oligosaccharides market — including pharmaceuticals but also cosmetics and other uses — reached US$ 3.56 billion in 2024 (Precedence Research, 2025). Both markets are growing rapidly, with marine-derived pharmaceuticals sales increasing by 7.5% between 2018 and 2022 (Fact.MR, 2023). However, access to marine genetic resources remains highly inequitable, with nearly half of all patents owned by a single company (Blasiak et al., 2018). Moreover, a critical point is the harvest of sufficient amounts of compounds without harming the marine environment (Lindequist, 2016).

**US$ 2.6 trillion ocean economy powers 134 million jobs** — but at an environmental cost. Between 1995 and 2020, the ocean economy contributed between 3% and 4% of total global gross value added, doubling from US$ 1.3 trillion to US$ 2.6 trillion (OECD, 2025). Its share of global employment also rose, from 3.5% to 4.7%, with 134 million full-time equivalent (FTE) jobs in 2019 before the COVID-19 pandemic. In 2022, 43.9 million people were engaged in marine fisheries, aquaculture, or unspecified subsectors (FAO, 2024). The expansion has been largely driven by two sectors with high environmental impacts: offshore oil and gas (US$ 988 billion in 2020) and coastal and marine tourism (US$ 789 billion in 2019), the latter employing 80 million FTEs. Offshore wind and marine renewable energies, however, experienced the fastest growth, with gross value added rising from US$ 38.2 million in 2000 to US$ 4.6 billion in 2020 (OECD, 2025; Jouffray et al., 2020). Reducing the environmental costs of ocean-based industries — through expanding alternative energy sources and improving sustainability — presents major opportunities for sustainable economic growth.

**Small-scale fisheries support 88% of marine harvest jobs and US$ 51.8 billion.** Marine small-scale fisheries account for a significant fraction of the total annual marine landings with an estimated 25.1 million tonnes (31.2% of global landings) (FAO et al., 2023; Gutierrez et al., 2023), representing an average (2013–2017) US$ 51.8 billion annual landed economic value, higher than the revenues of cruise tourism, port activities or offshore wind (Virdin et al., 2023). Although data remain incomplete and vary across regions, available estimates suggest that small-scale fisheries generate around 40% of global fisheries catches (Basurto et al., 2025). They employ approximately 12.9 million people in the harvesting segment, which represents 88.1% of total employment in marine fisheries value chains; they contribute directly to sustainable development by providing livelihoods, cultural value, and critical nutrition — especially to vulnerable societal groups (Basurto et al., 2025). Strengthening the sustainability and resilience of small-scale fisheries — including addressing pressures from other ocean sectors — is key to safeguarding their vital role in eradicating poverty, hunger and malnutrition.

## 4 Conclusions

With the growing demand for evidence-based information, there is an urgent need to improve regular access to coherent, cross-disciplinary ocean knowledge in order to foster well-informed awareness. The Starfish Barometer addresses this need by acting as a recurring synthesis and access point for such knowledge, drawing on multilateral scientific assessments — such as those from the IPCC, IPBES, and the World Ocean Assessment — as well as recent international reports and peer-reviewed academic research. Several global initiatives already contribute to monitoring various aspects of ocean health, such as the Ocean Health Index (Halpern et al., 2012, 2017) or the Ocean State Report (von Schuckmann et al., 2024). The Starfish Barometer complements these efforts by adopting a distinct narrative format and focusing each year on a carefully selected set of developments. Its symbolic structure and non-exhaustive nature allow it to highlight new evidence, figures, or issues as they emerge, helping connect scientific knowledge to a broader civic and policy-oriented conversation. Throughout the paper, we used Ocean capitalized, just as one does for Earth, to underscore the scientific recognition of a single, interconnected global Ocean. This usage aligns with the convention adopted by IPCC (2019).

Each year, the Starfish Barometer will spotlight a curated set of ocean-related developments that have stood out over the past twelve months. For its first edition, the Starfish Barometer has brought together a curated selection of ocean-related developments, chosen for their global relevance and scientific robustness, drawing on the most recent information available — even when published prior to the past year.

The view of this first Starfish Barometer reveals that the Ocean is undergoing rapid and alarming changes, marked by record heat, rising seas, widespread species decline, and threatened major ecosystems. Increasing pressures from rising greenhouse gas emissions and overexploitation are degrading the health of the Ocean, and pollution and habitat loss, particularly in coastal areas, are intensifying these challenges, while gaps in the global monitoring system hamper effective responses. The escalating costs of tropical storms and floods, coupled with rising migration fatalities, underscore the disproportionate burden for vulnerable populations and ecosystems. Likewise, the persistent threat of marine plastic pollution exposes the environmental injustice faced by both marine life and at-risk human communities, exacerbating existing vulnerabilities. At the same time, global ocean protection is advancing, with marine protected areas now covering 8.34% of the Ocean, but urgent action is still needed to reach common targets. Milestones such as the UN High Seas Treaty, progress on plastic pollution agreements, and a growing focus on equity and ocean literacy are driving significant momentum towards stronger and more sustainable ocean governance. Also, the ocean economy is experiencing significant growth, driven by sectors such as marine aquaculture, offshore wind, and tourism. However, this expansion also brings environmental challenges, with the continued pressure on marine ecosystems from high-demand industries. Small-scale fisheries continue to play a critical role in both global food production and coastal livelihoods, but are increasingly vulnerable to climate change and pressures from other ocean activities.

With the third United Nations Ocean Conference taking place in June, and the ratification of the High-Seas Treaty on the horizon, 2025 stands as a critical juncture for ocean governance — where multilateral cooperation and concrete actions are more urgent than ever. The 2025 Ocean Starfish Barometer highlights several strategic leverage points to accelerate progress on SDG 14 (UN, 2015), including reducing marine pollution (14.1), protecting ecosystems (14.2), tackling overfishing and

destructive practices (14.4), conserving marine areas (14.5), strengthening ocean science (14.A), supporting small-scale fishers (14.B), and upholding international law (14.C). It also highlights major gaps, such as the lack of an operational global monitoring system for plastic pollution in the Ocean.

These alignments demonstrate the Starfish Barometer's potential to serve not only as a communication tool, but also as a monitoring instrument that can contribute to tracking progress toward global ocean commitments. Beyond its global scope, the Starfish Barometer offers a flexible and adaptable framework — one that can be tailored for use at national or regional levels, integrated into educational programs, or used by institutions and NGOs to support reporting and awareness-raising. Its accessible design allows it to serve multiple purposes, from fostering civic engagement to informing policy dialogue. By connecting people to the Ocean through clear, meaningful insights, the Starfish Barometer seeks to spark reflection, responsibility, and action. It is a reminder that while the Ocean sustains life on Earth, its future — and ours — depends on the decisions we make today.

*Author contributions.* The concept of the Starfish Barometer was developed by M.L. and P.B., with the support of P.V.; M.L. provided overall leadership for this work, and led the writing of the manuscript with assistance from K.v.S; all coauthors contributed to content curation and manuscript preparation.

*Competing interests.* The authors declare having no competing interests.

*Acknowledgements.* The authors wish to warmly thank the international scientific committee of the One Ocean Science Congress (OOSC, Nice June 3–6 2025) namely Janine Adams, Tamatoa Bambridge, Sanae Chiba, Jorge Cortés-Núñez, Carlos Duarte, Stefan Gelcich, Jessica Gephart, Kristina Gjerde, Deborah Greaves, Peter Haugan, François Houllier, Jean-Pierre Gattuso, Daoji Li, Mere Takoko, Arthur Tuda, for agreeing to oversee this draft on very short notice. The authors also wish to express their special thanks to Thomas Froelicher and William Cheung, both members of the OOSC scientific committee, who kindly acted as reviewers outside the oversight group to ensure the neutrality and integrity of the review process. Their promptness, availability, and thoughtful comments were particularly appreciated. This paper would not have come to life without the dedication of Editor-in-Chief Marilaure Grégoire and the entire editorial staff, whose efficiency and commitment made it possible to complete the editorial process. The Starfish Barometer was developed with the support of Olivier Poivre d'Arvor, French Ambassador for the Ocean and the Poles, and the French President's Special Envoy for the United Nations Ocean Conference (UNOC3). Special thanks to the International Plateform for Ocean Sustainability (IPOS), for sowing the first seeds of the concept, and to CNRS, IFREMER and IRD for their scientific support. Several colleagues contributed to gathering the most up-to-date data, including Pierre Friedlingstein and Robbie Andrew from the Global Carbon project, Pierre Cariou and Laurent Fedi from Bordeaux Kedge Business School, Kira Coley from Ocean Census, Kensuke Obara from the Nippon foundation, Charles Loiseau from CRIOBE, and Florian Kirchner, Virginie Tsilibaris and Haizea Jimenez from IUCN. We also thank Laurence Collet for her contribution to the Starfish diagram and Clément Haëck for formatting the article and bibliography.

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
