# Peer review of "The 2025 Starfish Barometer"

_State of the Planet, 2025_

## Author Comment (AC1)

**Answer to reviews**

We warmly thank the editor in chief, Marilaure Grégoire, the editorial staff, and the reviewers for their careful reading, insightful comments, and constructive suggestions. We are especially grateful for their support and commitment in conducting this peer-review process under exceptionally tight timelines, and for helping us bring this inaugural edition of the Starfish Barometer to publication much faster than is customary.

We have carefully considered their suggestions, many of which called for methodological clarification. In response, we propose to introduce a key improvement: the addition of a dedicated "Methodology" section. This new section offers greater transparency about how the Barometer is constructed and how it will evolve over time, including the criteria used to select content and the editorial principles that guide its structure.

We first present this new Methodology section, as it answers several comments, and then provide point-by-point responses to each of the reviewers' comments.

This document presents our answers to both reviewers.

*Our answers are in italic,* **reviewers' comments are in bold**, copied original text is in black and our proposed additions to the text are in blue.

This is the new Methodology section that we propose to include after the Introduction section:

"**Methodology**. The content of the Starfish Barometer was curated by a multidisciplinary group of experts (the co-authors), who identified and selected significant developments to be featured in each of the five thematic branches. The process began with each expert proposing candidate news items for their respective branch, based on a set of common criteria: global relevance, robustness of the underlying data, significance of the development, and relevance to the Ocean-human relationship. Sources included scientific assessments (e.g. IPCC, FAO, OECD), new or updated datasets (e.g. Global Carbon Budget), recent peer-reviewed scientific publications, international policy developments, data gaps, and emerging alerts.

The proposed items were then collectively reviewed and discussed by the expert group to determine their relevance and final selection. This process is qualitative and adaptive by design, guided by transparent criteria and expert judgment to ensure that each Barometer edition remains accessible to the general public and focused on widely relevant issues. Each branch was limited to one page and to four key messages, anchored in recent developments and supported, in most cases, by figures and trend analyses.

The allocation of news items to specific branches is guided by thematic relevance but is not exclusive, as many ocean-related developments intersect multiple dimensions. For example, certain topics—such as IUU fishing—could be considered both a human pressure and a threat to societal equity. In such cases, items were assigned to the branch most closely aligned with the primary focus of the issue, following a consistent editorial framework rather than rigid topic boundaries.

We intentionally chose not to establish explicit logical connections between the news items within or across branches. While the branches together form a coherent picture of the Ocean-human relationship, forcing a linear or causal narrative might create an illusion of completeness

or determinism that does not reflect the complexity of ocean issues. Instead, each message stands on its own as a robust signal from the past year, contributing to a broader mosaic of understanding.

All figures presented in the Starfish Barometer are drawn directly from authoritative sources and are reported as published, in order to ensure transparency and traceability. Readers interested in the methodological details or associated uncertainties are encouraged to consult the original sources.

Each edition of the Starfish Barometer will follow the same symbolic and structural framework, with the five same thematic arms: Ocean state, human pressures, societal harms, protection efforts and opportunities for humanity. However, the content within each arm will evolve yearly to reflect the most recent and relevant developments. When new data become available—such as for carbon emissions or sea level—they will be updated accordingly. Yet not all information is refreshed annually, and the Barometer's non-exhaustive approach provides the necessary flexibility to highlight either new figures or important issues that may vary from one year to the next. In this sense, the Barometer differs from initiatives like the Global Carbon Project (Friedlingstein et al., 2025) or the Ocean health index (Halpern et al., 2012): rather than systematically updating a fixed dataset, it offers a curated, narrative-based selection that captures the most meaningful signals of the year."

**Reviewer 1**

The main aim of this paper is to introduce the newly developed Starfish Barometer, which will be updated and published annually at the World Ocean Day. The Barometer is intended to provide a comprehensive yet accessible summary of the current state of the ocean, combining scientific evidence with a format that is approachable to the general public. Using the symbolic framework of a starfish, the Barometer addresses five key thematic areas: the condition of the ocean, human pressures, societal harms, protective efforts, and opportunities for humanity. The 2025 edition highlights both persistent challenges and notable progress.

I find the concept of the Starfish Barometer highly commendable. Its focus on a selected number of key issues each year—rather than attempting an exhaustive overview—is both pragmatic and effective. While the content is based on existing scientific literature and reports, the Barometer succeeds in synthesizing the complex and fragmented data across multiple dimensions of ocean health into a clear and engaging narrative.

I have outlined below a set of minor but important comments that should be addressed prior to recommending this paper for publication. I look forward to seeing this inaugural edition of the Starfish Barometer published soon.

We sincerely thank Reviewer 1 for their thoughtful and encouraging feedback on the Starfish Barometer. We are grateful for the positive reception of both the concept and the structure of the Barometer, as well as for the constructive comments aimed at strengthening the clarity and impact of this inaugural edition. We have carefully considered all suggestions and we propose to implement the necessary changes or clarifications as outlined in the responses below.

**Minor Comments and Suggestions**

**General:**

It remains unclear whether the paper itself will be updated annually, or if only the data and visualizations on the accompanying website will be revised each year. Is the intention to maintain a 'living document'—similar to the Global Carbon Project (GCP) that evolves over time? I strongly recommend implementing such a model to ensure transparency and traceability across editions.

Our intention is indeed to publish a new Starfish Barometer paper each year, with a consistent structure anchored in the framework of the five arms. While the overarching structure and the five thematic areas will remain unchanged, the content within each arm will evolve annually, reflecting newly available data and emerging issues.

As with the Global Carbon Project, we will update indicators such as carbon emissions when new numbers are released. However, not all metrics are updated every year. The nonexhaustive nature of the Barometer gives us the flexibility to incorporate new or alternative information each year, depending on availability and relevance. This adaptability is a strength of the format, allowing us to highlight key developments without being constrained to a fixed set of indicators. In this respect, the Starfish Barometer is somewhat different from the GCP, offering a more narrative and selectively curated perspective.

We have clarified this intention in the revised manuscript by adding a new "Methodology section", and we will ensure that all editions are clearly dated, archived, and accessible for full transparency and traceability over time. Please see above for a full description of the content of this new Methodology section.

**Abstract:**

Given that the ocean reached record-high surface temperatures in 2023 and 2024, it would strengthen the abstract to mention this explicitly, as it contextualizes the urgency of the findings.

Thank you for this advice, and we propose to add the wording accordingly: '2024 ocean temperatures break the 64-year record as marine heatwaves show a marked increase globally'.

The wording of 'as marine heatwaves show a marked increase globally' will be added in the abstract as well to ensure that there is no miscommunication with respect to this signal (long-term vs natural/extreme) such as published in Teerhar et al., 2025.

Line 16: The phrase "only 8.34%" introduces a value judgment. I suggest removing "only" to maintain a more objective tone appropriate for the abstract.

Agreed, we will remove "only"

Introduction:

It may be useful to acknowledge that the selection of focus areas—represented by the starfish arms—inevitably involves some degree of subjectivity. Stating this openly would reinforce the transparency of the framework.

Thank you for this suggestion. We acknowledge that the curation process is "guided by transparent criteria and expert judgment" in the new Methodology section.

Section 2.1:

Consider including more detail on the record increase in global sea surface temperature in 2023/2024, which surpassed the 2015/16 record by approximately 0.25°C. This was a very unusual but not entirely unexpected event. While El Niño contributed, it was not sufficient to explain the jump alone. Terhaar et al. (2025, Nature) provide evidence countering claims that current models fail to capture such changes. Including this reference would strengthen the section.

Thank you for raising this aspect, and referring to the most recent literature on this topic. We propose to revise the text and references accordingly:

"2024 marks a year of intense, persistent and widespread heatwaves in the ocean, and ocean heat content reached its highest level in the 64 years for which we have reliable recorded global observations (since 1960), surpassing the previous record high set in 2023 (WMO, 2025; Cheng et al., 2025; Pan et al., 2025), and the 2015/16 record by 0.25°C at the ocean surface (Terhaar et al., 2025). The record ocean surface temperature values reflect natural variability amplified by long-term global warming—an event unlikely to occur without the underlying climate trend (Terhaar et al., 2025; Guinaldo et al., 2025)."

Terhaar, J., Burger, F. A., Vogt, L., Frölicher, T. L., & Stocker, T. F. (2025). Record sea surface temperature jump in 2023–2024 unlikely but not unexpected. *Nature*, *639*(8056), 942–946. https://doi.org/10.1038/s41586-025-08674-z

Guinaldo, T., Cassou, C., Sallée, J.-B., & Liné, A. (2025). Internal variability effect doped by climate change drove the 2023 marine heat extreme in the North Atlantic. *Communications Earth & Environment*, 6(1), 291. https://doi.org/10.1038/s43247-025-02197-1

Clarify the term "acidity" or "acidification." If the reported 30% increase refers to  $[H^+]$  concentration, this should be stated explicitly to avoid confusion with other metrics such as changes in aragonite saturation ( $\Omega$ ).

Thank you for the comment and we propose to complete the sentence:

"Ocean surface acidity (in [H+] concentration) has risen by 30%"

More clearly link the fourth global coral bleaching event to ocean warming, as this connection is currently implied but not explicitly stated.

Thank you for this comment, we propose the following addition to the text:

"Following extreme conditions in 2023 driven by historically high heat stress, the 2024 coral bleaching event is the fourth global event on record since 1985 and the second in the past decade."

Figure 1:

The symbols on the individual arms of the starfish appear meaningful. Will these themes (i.e. symbols) serve as a recurring structure in future Barometer editions, or are they illustrative only? Clarification would help readers understand the role of this visual design.

Indeed, the 5 themes will serve as recurring figures. This is stated more clearly in the Methodology section.

Section 2.2:

Consider mentioning that international shipping emissions contribute approximately 1.6% of total global fossil fuel  $CO_2$  emissions. While this share is relatively small, the localized health and environmental impacts—particularly near ports and coastal communities—are significant and not currently discussed. Including these could enhance the 'Societal Harms' section/paragraph.

Thank you for this suggestion, we have rephrased the end of section 2.2 to insist on the relatively small contribution:

"Emissions from international shipping are also on the rise, with a projected 0.6 billion tons of CO2 in 2024 (representing 1.6% of total global emissions), a 2.7% increase compared to 2023."

Thank you also for your suggestion to discuss the health impacts near ports and for coastal communities as part of the Societal Harms. We do not currently have all the necessary information to include it in the 2025 edition, but we will definitely keep it in mind for the 2026 edition.

**The figure of 37.7% for unsustainable fishing may appear overly precise. If the underlying uncertainty is substantial, consider rounding or contextualizing the number. This comment also applies to other numbers in the text.**

Thank you for this comment. We acknowledge the concern regarding the apparent precision of figures such as the 37.7% of overexploited fish stocks. However, our intention is to faithfully reflect the official numbers reported by primary sources—in this case, the 2024 FAO report—which presents this figure without a stated uncertainty range. Discussing the uncertainties behind each estimate would substantially increase the length and complexity of the paper, potentially diluting the clarity and accessibility of our key messages. We have therefore chosen to report the figures as published, while referencing the original sources to allow interested readers to explore the underlying methodologies and uncertainties in more detail. We now explain this position in the new methodology section.

**The phrase "dramatic rise in nearshore macroalgal blooms" would benefit from quantification. What constitutes "dramatic" in this context—10%, 100%, more?**

Thank you for your comment. There is currently no comprehensive global estimate of the magnitude of this increase, only mentioning that the frequency and extent of green tides have dramatically increased since 2000, with more and more reporting of such events across various regions. We propose to remove the word "dramatic" and to include some extra information:

"Climate change and pollution-driven eutrophication have contributed to a dramatic rise in the reporting of nearshore macroalgal blooms (green tides) over the past two decades (Ren et al., 2024), worldwide, although comprehensive global estimates are lacking."

**Section 2.3:**

It is unclear whether the reported loss of life relates to ocean-induced migration (e.g., from sea-level rise or climate variability) or to other causes. If data are available on total ocean-related mortality, it may be valuable to include here.

Thank you for your comment. While we acknowledge that environmental factors are important drivers of migration, the exact relationship between oceanic changes and migrant fatalities remains uncertain and requires further investigation. Unfortunately, we do not have specific data on ocean-related mortality. We propose to add a comment at the end of the paragraph to acknowledge that point:

"The specific relationship between oceanic changes (e.g., from sea-level rise, depletion of marine resources due to overfishing or climate variability) and migrant fatalities remains unclear and would require further investigation"

**Section 2.4:**

While the text notes that MPAs expanded by only 0.007% between 2023 and 2024, it would be helpful to include figures showing the change over a longer timescale (e.g., the past decade) to give readers a sense of long-term progress.

Thank you for your suggestion. We propose to modify the first sentence and add one reference to reflect the longer timescale:

"Over the past decade, the global coverage of MPAs has increased significantly, from 3.72% in 2015 (Lubchenco and Grorud-Colvert, 2015) to 8.34% (corresponding to 30.238 millions of km2) in 2024 (protected-planet, 2025), but still far from the 30x30 target (protecting 30% of the coastal and marine waters by 2030) (unep-wcmc, 2024)."

Lubchenco, J., & Grorud-Colvert, K. (2015). Making waves: The science and politics of ocean protection. *Science*, *350*(6259), 382–383. https://doi.org/10.1126/science.aad5443

Although I am not an expert in ocean governance, I believe the International Tribunal for the Law of the Sea (ITLOS) recently ruled that  $CO_2$  emissions constitute marine

pollution under UNCLOS, requiring states to take preventative measures. This is a significant legal development and should be mentioned in this section.

Thank you for your suggestion. We propose to add this sentence and associated reference:

"In 2024, the International Tribunal for the Law of the Sea issued a landmark opinion (ITLOS, 2024), addressing States' obligations to protect the oceans from climate change impacts within the framework of the United Nations Convention on the Law of the Sea (UNCLOS)."

ITLOS, 2024: https://www.itlos.org/fileadmin/itlos/documents/cases/31/Advisory Opinion/C31 Adv Op 21. 05.2024 orig.pdf

**Reviewer 2**

The manuscript presents a timely and important initiative which seeks to assess and communicate the status and trends of ocean health and related societal dimensions. The concept is commendable and the presentation is generally clear. My understanding is that this is intended for non-technical readers. However, several areas require further clarification, elaboration, and stronger connection between sections to improve coherence and rigor.

I provide below specific comments and suggestions for improvement directly on the manuscript.

We thank Reviewer 2 for their constructive feedback. We greatly appreciate the recognition of the relevance and importance of the Starfish Barometer initiative, as well as the encouragement regarding its value for non-technical audiences. We also thank the reviewer for identifying areas where further clarification could improve the manuscript.

**Major Comments**

**1. Intended Audience and Positioning**

• It would be beneficial for the authors to clearly indicate the main audience of the initiative early in the manuscript. Identifying whether the intended readership includes policymakers, the general public, the scientific community, or all of the above will help tailor the messaging and focus (both in the abstract and in the introduction)

We thank the reviewer for this helpful suggestion. We agree that clarifying the primary audience is essential to contextualize the purpose and format of the Barometer. We propose to make this explicit in both the abstract and the introduction (second paragraph), indicating that the Starfish Barometer is primarily intended for a broad, non-specialist audience—including policymakers, educators, civil society, and the general public—while remaining grounded in peer-reviewed science to ensure credibility and usefulness for the scientific community as well:

Abstract: "global significance and grounded in the most up-to-date scientific evidence, intended for a broad, non-specialist audience"

Introduction: "Its strength lies in how it brings together this dispersed knowledge, synthesizing it into a comprehensive and integrated overview of the Ocean, aligned with the United Nations Sustainable Development Goal related to the Ocean (SDG 14; UN, 2015). The Barometer is designed to serve a broad, non-specialist audience—including policymakers, educators, civil

society actors, and the general public—while remaining grounded in peer-reviewed science to ensure credibility and relevance for the scientific community as well."

**• Introduction: I suggest acknowledging similar existing initiatives (e.g., Ocean Health Index) and clearly articulating how the Starfish Barometer differentiates itself in scope, methodology, or application.**

We thank the reviewer for this suggestion. Rather than expanding the introduction, we have chosen to integrate this articulation into the conclusion, where we now acknowledge complementary initiatives such as the Ocean Health Index and the Ocean State Report. This allows us to situate the Starfish Barometer within the broader landscape of ocean monitoring and communication efforts, while emphasizing its distinctive approach in terms of narrative framing, selective annual curation, and audience engagement:

"The Starfish Barometer addresses this need by acting as a recurring synthesis and access point for such knowledge, drawing on multilateral scientific assessments — such as those from the IPCC, IPBES, and the World Ocean Assessment—as well as recent international reports and peer-reviewed academic research. Several global initiatives already contribute to monitoring various aspects of ocean health, such as the Ocean Health Index (Halpern et al., 2012, 2017) or the Ocean State Report (von Schuckmann et al., 2024). The Starfish Barometer complements these efforts by adopting a distinct narrative format and focusing each year on a carefully selected set of developments. Its symbolic structure and non-exhaustive nature allow it to highlight new evidence, figures, or issues as they emerge—helping connect scientific knowledge to a broader civic and policy-oriented conversation."

Halpern, B., Longo, C., Hardy, D. et al. An index to assess the health and benefits of the global ocean. Nature 488, 615–620 (2012). https://doi.org/10.1038/nature11397

Halpern, B. S., Frazier, M., Afflerbach, J., O'Hara, C., Katona, S., Lowndes, J. S. S., et al. (2017). Drivers and implications of change in global ocean health over the past five years. PLoS ONE, 12(7), e0178267. https://doi.org/10.1371/journal.pone.0178267

**2. Framework and Rationale**

• A paragraph explaining the rationale behind the choice of focal topics for 2025 should be included. If the focal topics are intended to change with each assessment cycle, this adaptive approach should be explained

• A brief summary of the assessment methods should be presented in the main text, with detailed methodologies referenced in the supplementary materials.

We thank the reviewer for these two important suggestions. We have added a new section to the revised manuscript clarifying the rationale and process behind the selection of focal topics for the 2025 edition. The curation process began with the formation of a multidisciplinary group of experts (the co-authors), each invited to propose relevant news items for one of the five branches of the Barometer, based on defined selection criteria. These included: global scope, scientific robustness, relevance to the Ocean-human relationship, and significance of recent developments.

In a second step, the expert group collectively reviewed and discussed the proposed items to determine their final inclusion. This evaluation was intentionally qualitative and guided by two objectives: (1) ensuring the content remains accessible and engaging to a broad, non-specialist audience, and (2) highlighting ocean-related themes that are both timely and of

general public interest. Each branch was limited to one page and structured around three to four key messages, built from news items and contextualized with available figures and trends. A summary of this approach is now included in the manuscript, in a new methodology section.

**3. Scientific Accuracy and Completeness**

**• Line 60: When discussing ocean warming, providing an estimated rate of warming would strengthen the context.**

Thank you for your suggestion. While we agree that including a quantified rate of ocean warming  $(0.6 \pm 0.1 \text{ W} \cdot \text{m-2} \text{ since } 1971, \text{ WMO}, 2025)$  could offer useful context, we chose not to do so in this case in order to preserve accessibility for a general audience.

**• Line 69: Statements about changing risk levels of species could be strengthened by quantifying both the number of species increasing in risk and those decreasing, as available from the Red List Index for marine species.**

**Thank you for your suggestion. We propose to add this information:**

"As of the most recent update (May 2025), 1,677 marine species are listed as at risk of extinction on the IUCN Red List, including 291 classified as critically endangered, 647 as endangered, and 739 as vulnerable (IUCN, 2025). Among these, 1,211 species are experiencing declining populations, 28 are increasing, and the remainder are either stable or have unknown trends."

**• Line 74: The impacts on deep-sea ecosystems can mention multiple stressors, including climate change, deep-sea mining, and bottom trawling.**

Thank you and we propose to add a sentence to reflect that, as well as a new reference:

"This pace of discovery highlights a critical risk: countless marine species may vanish before science even has a chance to identify them, particularly in the deep-sea where multiple stressors puts them at risk (climate change, deep-sea mining and bottom trawling, Global Deep Sea Consultation, 2025) making conservation efforts all the more urgent (Ocean Census, 2025)"

**Global Deep-sea consultation, 2025: https://ipos.earth/global-deep-sea-consultation-pilot-project**

**• Line 78: The discussion on extreme conditions should briefly define or exemplify these conditions**

Thank you, and according to the first reviewer, we propose to change the text to:

"Following extreme conditions in 2023 driven by historically high heat stress, the 2024 coral bleaching event is the fourth global event on record since 1985 and the second in the past decade."

**4. Linking Drivers and Pressures**

• Line 84-87: The manuscript should more clearly explain the linkages between CO2 emissions and associated oceanic pressures, including climate change and acidification.

Thank you, this was indeed lacking. We propose:

"Fossil fuel  $CO_2$  emissions are the main contributor to recent climate change (IPCC, 2021). They also lead to ocean acidification as approximately one-third of these emissions are absorbed by the ocean (Friedlingstein et al., 2025)."

• Line 88-89: The use of "sustainable level" is currently narrow, defined primarily as maximum sustainable yield (MSY). It would be helpful to acknowledge that fisheries can be unsustainable even at MSY levels due to ecological complexities such as trophic disruptions and bycatch.

Thank you for this useful suggestion, we have modified the paragraph accordingly:

"The percentage of marine stocks fished at unsustainable levels (i.e. beyond Maximum Sustainable Yield) has increased since the mid-1970s, from 10% in 1974 to 37.7% in 2021 (FAO, 2024). This estimate represents only part of the unsustainability challenge, as it does not account for broader ecological and social impacts of fisheries (Asche et al., 2025), nor for unreported and illegal fishing ."

Asche, F., Garlock, T. M., Anderson, J. L., Pincinato, R. B., Anderson, C. M., Camp, E. V., et al. (2025). A Review of Global Fisheries Performance. *Fish and Fisheries*, *26*(3), 444–453. https://doi.org/10.1111/faf.12890

**• Line 103-104: It would be useful to briefly mention the main pathways of impacts here - through pollution, habitat destruction, increased risks of marine pathogens, parasites and invasive species.**

**Thank you for the suggestion. We propose:**

"The multiple uses of coastal areas — like coastal infrastructures, tourism, fish farming, and offshore structures — are putting more pressure on ocean habitats, through pollution, habitat destruction, increased risks of marine pathogens, parasites and invasive species, especially as coastal populations grow"

**5. Integration Across Sections**

• The manuscript could benefit from integrating between sections. Currently, the selection topics in 2.2, 2.3 and 2.4 does not fully align. For example, E.g., 2.3 does not have any mention of the impacts of over-exploitation or the lost of coast habitats on the society - topics that are highlighted in 2.2. Also, pollution focuses on plastic only, but coastal hypoxia is discussed in 2.2, much of which is driven by excessive nutrients, leading to coastal hypoxia. It would be useful to discuss the implications of the pressures highlighted in 2.2 for the society in 2.3. Similarly, it would be useful to strengthen the linkages between the responses (2.4) to 2.2 and 2.3.

We fully agree that the different topics within each arm do not align from one arm to the other. This separation is the result of a deliberate editorial choice, which we have now clarified in the new methodology section. In curating the Barometer, the expert group intentionally avoided constructing explicit causal or logical links between items or across branches. This decision was motivated by two considerations: (1) to preserve the clarity and self-sufficiency of each branch, and (2) to avoid suggesting a false sense of completeness or linearity in the complex ocean-human interactions.

**6. Others**

**• Efforts to address pollution and climate change should be explicitly mentioned in the management response sections.**

We recall here that the intention of the Barometer is not to be exhaustive. Not all stressors, responses, or impacts can be covered in each edition. The Barometer highlights a selection of developments that are particularly relevant for the year in question. While not all pollution or climate-related efforts are included in the current edition, the methodology remains flexible and open to featuring such efforts in future editions.

**• LineL 199-207: The discussion of small-scale fisheries should also emphasize their importance in promoting equity and sustainability. They provide livelihoods, cultural value, and critical nutrition, especially for vulnerable societal groups, in contrast to large-scale industrial fisheries. Generally, they contributes more to sustainable development.**

We agree that the role of small-scale fisheries in promoting equity and sustainability deserves greater emphasis. We propose to revise the paragraph accordingly:

"Small-scale fisheries..... They employ approximately 12.9 million people in the harvesting segment, which represents 88.1% of total employment in marine fisheries value chains; they contribute directly to sustainable development by providing livelihoods, cultural value, and critical nutrition—especially to vulnerable societal groups (Basurto et al., 2025). Strengthening the sustainability and resilience of small-scale fisheries—including addressing pressures from other ocean sectors—is key to safeguarding their vital role in eradicating poverty, hunger and malnutrition in economy and food security."

**• The anticipated benefits of marine protected areas (MPAs) should be clearly articulated.**

We agree, and we propose to conclude the news on MPAs with the following addition:

"Ensuring that the global network of MPAs delivers its expected benefits for climate, biodiversity, and food security requires not only achieving the 30% coverage target, but also increasing the level of protection within MPAs (Arneth et al., 2023).

Arneth, A., Leadley, P., Claudet, J., Coll, M., Rondinini, C., Rounsevell, M. D. A., et al. (2023). Making protected areas effective for biodiversity, climate and food. *Global Change Biology*, *29*(14), 3883–3894. https://doi.org/10.1111/gcb.16664

**• Line 180: A note on the importance of ensuring sustainability in marine biological products would also strengthen the narrative.**

We thank the reviewer for this remark. We propose to end the news on the Marine-health market with the following addition:

"Moreover, a critical point is the harvest of sufficient amounts of compounds without harming the marine environment (Lindequist, 2016)".

Lindequist, U. (2016). Marine-Derived Pharmaceuticals – Challenges and Opportunities. *Biomolecules & Therapeutics*, 24(6), 561–571. https://doi.org/10.4062/biomolther.2016.181

**Minor Comments**

**• Line 6: Phrases like "ocean-related developments" need further clarification. It is important to define such terms to avoid ambiguity for readers from diverse backgrounds.**

Thank you for this helpful suggestion. We propose to add the following definition in the introduction:

"Each year, the Starfish Barometer selects and curates a set of ocean-related developments—such as new or updated scientific findings, international policy decisions, or governance milestones—chosen for their global relevance and based on the most recent knowledge available at the time of publication."

**• Line 29: Suggest rephrasing as latest availability information on historical changes and current status and trends, instead of "facts".. currently, it makes "predicted futures" sound unscientific.**

Thank you for this suggestion, we propose to reformulate the initial sentence:

"The Starfish Barometer takes stock of the latest available facts information on historical changes and current status and trends, rather than focusing on predicted projected futures".

**• Line 97: Specific terms such as "struggling" should be clarified (e.g., does it mean species are being impacted?).**

We propose to replace "struggling" by "showing increasing signs of degradation"

---

## Author Response (AR2)

May, 13th 2025

Dear Marilaure Grégoire,

Thank you for recommending our work, *"The 2025 Starfish Barometer"*, for publication in State of the Planet.

We have addressed the two minor comments as follows (added text in blue):

1) "2024 ocean temperatures break the 64-year record with sea-surface temperature and  marine heatwaves showing a marked increase globally"
2) habitat destruction

Many thanks again for handling the editorial process,

Kind regards,

Marina Lévy